# Role of Vitamin D in Preventing and Treating Selected Extraskeletal Diseases—An Umbrella Review

**DOI:** 10.3390/nu12040969

**Published:** 2020-03-31

**Authors:** Friederike Maretzke, Angela Bechthold, Sarah Egert, Jana B. Ernst, Debora Melo van Lent, Stefan Pilz, Jörg Reichrath, Gabriele I. Stangl, Peter Stehle, Dorothee Volkert, Michael Wagner, Julia Waizenegger, Armin Zittermann, Jakob Linseisen

**Affiliations:** 1German Nutrition Society, 53175 Bonn, Germany; rikimaretzke@hotmail.com (F.M.); bechthold@dge.de (A.B.); j-b-e@gmx.net (J.B.E.); j.linseisen@unika-t.de (J.L.); 2Institute of Nutritional Medicine, University of Hohenheim, 70599 Stuttgart, Germany; sarah.egert@uni-hohenheim.de; 3Glenn Biggs Institute for Alzheimer’s and Neurodegenerative Diseases, University of Texas Health Sciences Center, San Antonio, TX 78229, USA; deboramelovanlentresearch@gmail.com; 4Division of Endocrinology and Diabetology, Department of Internal Medicine, Medical University of Graz, 8036 Graz, Austria; stefan.pilz@medunigraz.at; 5Department of Adult and Pediatric Dermatology, Venereology, Allergology, University Hospital Saarland, 66424 Homburg, Germany; joerg.reichrath@uks.eu; 6Institute for Agricultural and Nutritional Sciences, Martin Luther University Halle-Wittenberg, 06120 Halle (Saale), Germany; gabriele.stangl@landw.uni-halle.de; 7Department of Nutrition and Food Sciences, University of Bonn, 53115 Bonn, Germany; pstehle@uni-bonn.de; 8Institute for Biomedicine of Aging, Friedrich-Alexander-Universität Erlangen-Nürnberg, 90408 Nuremberg, Germany; dorothee.volkert@fau.de; 9Department for Neurodegenerative Diseases and Geriatric Psychiatry, University Hospital Bonn, 53127 Bonn, Germany; michael.wagner@dzne.de; 10Clinic for Thoracic and Cardiovascular Surgery, Heart and Diabetes Center North Rhine-Westphalia, 32545 Bad Oeynhausen, Germany; azittermann@hdz-nrw.de; 11University Center of Health Sciences at Klinikum Augsburg (UNIKA-T), Ludwig Maximilian University of Munich, 86156 Augsburg, Germany

**Keywords:** vitamin D, 25-hydroxyvitamin D (25(OH)D), asthma, COPD, ARI, dementia and cognitive decline, depression, multiple sclerosis, T1DM, umbrella review

## Abstract

Evidence is accumulating that vitamin D may have beneficial effects on respiratory tract, autoimmune, neuro-degenerative, and mental diseases. The present umbrella review of systematic reviews (SRs) of cohort studies and randomised controlled trials (RCTs), plus single Mendelian randomisation studies aims to update current knowledge on the potential role of vitamin D in preventing and treating these extraskeletal diseases. Altogether, 73 SRs were identified. Observational data on primary prevention suggest an inverse association between vitamin D status and the risk of acute respiratory tract infections (ARI), dementia and cognitive decline, and depression, whereas studies regarding asthma, multiple sclerosis (MS), and type 1 diabetes mellitus (T1DM) are scarce. SRs of RCTs support observational data only for the risk of ARI. No respective RCTs are available for the prevention of chronic obstructive pulmonary disease (COPD), MS, and T1DM. SRs of RCTs indicate beneficial therapeutic effects in vitamin D-deficient patients with asthma and COPD, while effects on major depression and T1DM need to be further elucidated. Mendelian randomisation studies do not consistently support the results of SRs. Since several limitations of the included SRs and existing RCTs do not permit definitive conclusions regarding vitamin D and the selected diseases, further high-quality RCTs are warranted.

## 1. Introduction

Vitamin D occupies a unique position among vitamins as humans can meet their requirements via dual sources: through the consumption of vitamin D (supplements or vitamin D-fortified foods), as well as through formation in the human body via exposure of the skin to solar or artificial ultraviolet B (UVB) radiation. According to current guidelines, vitamin D is a conditionally indispensable nutrient: in situations where endogenous synthesis is strongly impaired (vulnerable person groups include infants, veiled or dark-skinned people, and home-bound older people such as nursing home residents [1]), age-specific requirements can be met with a daily vitamin D intake (supplement) of 10–20 µg (400–800 international units (IU)) [2,3,4,5]. The cutaneous vitamin D synthesis can provide approximately 25 µg (1000 IU) vitamin D per day being formed within only one minute of whole-body exposure at cloudless midlatitudes and solar noon (end of June at northern latitudes) [6]. Thus, under optimal physiological conditions, the endogenous synthesis is generally sufficient to maintain an adequate vitamin D status during the summer months. Regarding cutaneous vitamin D synthesis, it should be noted that in countries with high UV exposure, individual behaviour with avoidance of sun exposure (e.g., in Saudi Arabia) can lead to a relatively high prevalence of vitamin D deficiency, whereas above 37°N latitude there are marked decreases (80–100%, depending on latitude) in the number of UVB photons reaching the earth’s surface during the months of November through February. Hence, during winter the endogenous synthesis of vitamin D is limited or even negligible, thus, individuals are at a higher risk to develop vitamin D insufficiency [7]. Due to varying geographical latitudes and outdoor activities, the general population in Central Europe and North America synthesises merely 12.5–15 µg (500–600 IU) per day in summer and virtually no vitamin D in winter [1]. Naturally, only few foods contain vitamin D, including fatty fish (e.g., herring, mackerel), liver and fat from aquatic mammals (e.g., seals and polar bears), and egg yolks. The daily dietary intake in adults living in Central Europe is only about 2–3 µg (80–120 IU) [4,5]. 

Circulating 25-hydroxyvitamin D (25(OH)D) reflects both endogenous synthesis and dietary intake of vitamin D and is thus the internationally accepted marker for assessing vitamin D status [8,9]. While deficient vitamin D supply is predominantly defined as 25(OH)D serum concentrations below 25–30 nmol/l (10–12 ng/mL), there is no clear agreement on the optimal range [10,11]. The North American Institute of Medicine (IOM), the D-A-CH nutrition societies (D-A-CH: Germany, Austria, Switzerland), the Scandinavian nutrition societies, the German Osteology governing body (DVO) and the European Society for Clinic and Economic Aspects of Osteoporosis and Osteoarthritis all specify a serum 25(OH)D level of ≥50 nmol/l (20 ng/mL) as the lower target value for an adequate vitamin D supply. In contrast, the Endocrine Society and the International Osteoporosis Foundation consider an adequate vitamin D supply to be guaranteed at levels of at least 75 nmol/l (30 ng/mL). Despite inconsistent target values, there is a broad consensus that blood levels of 25(OH)D should not fall below 50 nmol/l [1,10,11]. Globally, a high prevalence of vitamin D deficiency has been reported [12,13,14]. Results of an analysis of 14 population studies throughout Europe showed that 13% of 55,844 European individuals had serum 25(OH)D concentrations of <30 nmol/l (12 ng/mL) and 40.4% concentrations below <50 nmol/l (20 ng/mL), irrespective of age group, ethnic mix, and latitude of study populations [12].

It has long been proven that the biologically most active vitamin D metabolite, 1,25(OH)_2_D (calcitriol), plays a pivotal role in regulating calcium and phosphate homoeostasis and is thus important for bone health [10,15]. Since calcitriol is a well-known ligand of the vitamin D receptor (VDR), which is a transcription factor that influences the expression of thousands of genes [16], it can thus be assumed that bioactive vitamin D has multiple other functions besides its role in mineral metabolism and skeletal health. The VDR and the enzyme 1-α-hydroxylase, of which the latter is necessary for the hydroxylation of 25(OH)D into calcitriol, are expressed in various types of body cells (e.g., renal proximal tubule cells, intestinal cells, keratinocytes, monocytes, T lymphocytes, and dendritic cells) [17]. In fact, abundant evidence exists concerning a potential association of vitamin D status or supplementation on cardiovascular diseases [18,19,20], cancer [21,22,23], type 2 diabetes mellitus [24,25], and hypertension [26,27]. Concerning cancer, a recently published MA of RCTs showed that vitamin D supplementation significantly reduced total cancer mortality but did not reduce total cancer incidence [28]. In addition, vitamin D may also be relevant for other autoimmune diseases such as rheumatoid arthritis [29]. Nevertheless, in a literature review of the German Nutrition Society on vitamin D and prevention of selected chronic diseases in 2011, the evidence for a causal relationship between vitamin D status and the reviewed diseases (cancer, type 2 diabetes mellitus, hypertension, cardiovascular diseases) was considered only ‘possible’ or ‘insufficient’ [10]. In line with this conclusion, two previously published umbrella reviews of meta-analyses (MAs) and systematic reviews (SRs) reported predominantly no clearly established effect of vitamin D supplementation on the risk of these diseases [30,31]. 

However, evidence is accumulating that vitamin D may have beneficial effects regarding the risk of neurodegenerative and mental diseases [18], as well as autoimmune [32] and respiratory diseases [33]. In this context, several RCTs and cohort studies, as well as SRs, have been published over the last few years, which were not considered in earlier umbrella reviews [30,31]. Therefore, the aim of this umbrella review is to provide a comprehensive update on the potential relationship between vitamin D supply and the prevention and therapy of the following specific extraskeletal diseases, which have not yet been comprehensively covered in previous reviews: asthma, COPD, ARI, dementia and cognitive decline, depression, MS, and T1DM. The rationale for such a review is further supported by the fact that there is an increase in uncritical self-supplementation of vitamin D in the general population that may be driven by unproven health claims of vitamin D in terms of certain extraskeletal health outcomes [34].

## 2. Methods 

### 2.1. Protocol, Registration, and Study Design

A prospectively developed methodological approach for this umbrella review was registered on PROSPERO (CRD42019103670). The present umbrella review comprises SRs (with or without MAs) of cohort studies and RCTs investigating the association between 25(OH)D status or vitamin D supplementation and the following extraskeletal health diseases (prevention and therapy): respiratory tract diseases (asthma, chronic obstructive pulmonary disease (COPD), and acute respiratory tract infections (ARI)), neurodegenerative and mental diseases (depression, dementia, and cognitive decline) and autoimmune diseases (multiple sclerosis (MS) and type 1 diabetes mellitus (T1DM)). In addition, relevant Mendelian randomisation studies were identified to complement the results of SRs. Mendelian randomisation uses genetic variants to study putative causal effects of modifiable exposures on an outcome. The principle behind is that if a biomarker, such as 25(OH)D, is involved in disease etiology, then the genetic factors influencing the biomarker will in turn influence disease risk. This method avoids some of the limitations of classical epidemiology (e.g., less prone to confounding, free of reverse causation) but has also its restrictions (see “Overall Discussion and Conclusion”, page 23, line 1071) [35,36].

### 2.2. Search Strategy and Eligibility Criteria

Systematic literature searches for SRs and MAs across the databases PubMed and Cochrane Reviews library, using comprehensive search strategies, were performed for each extraskeletal health outcome separately in January 2019 (dementia and cognitive decline, depression, COPD), March 2019 (asthma) and April 2019 (MS, T1DM, ARI). Search terms for vitamin D, study type (SR or MA) and the respective disease were combined. The search strategy is presented in Appendix A. 

All searches were restricted to articles in English or German language published after 01 January, 2010. Only SRs (with or without MAs) of at least two cohort studies and RCTs with human study participants were eligible. In general, only studies in adult populations were included, except for the endpoints asthma, ARI and T1DM which are also particularly significant in the early life. We aimed to include all study data, irrespective of the vitamin D application dose. However, as there is existing data suggesting that intermittent high bolus doses of vitamin D may either be not effective or cause adverse effects, in our work we payed particular attention to certain vitamin D dosing regimens when interpreting and discussing the results of included reviews for the respective outcomes. Titles and/or abstracts of retrieved studies were screened by at least two review authors according to pre-defined inclusion/exclusion criteria to identify potentially eligible studies (see Table 1). The full texts of these publications were retrieved and assessed for eligibility. Additionally, non-systematic literature searches across PubMed were performed to identify relevant Mendelian randomisation studies (search period: January–May 2019) investigating the association between vitamin D status and the respective health outcomes.

### 2.3. Data Extraction

Data from all included studies were extracted into a standardised form. The following data were extracted: first author, year of publication, study type, study period, number of included studies, study population (number, gender, and age), exposure(s), endpoint(s)/outcome, effect estimates, *p*-values, heterogeneity estimates and subgroup analyses.

### 2.4. Assessment of Methodological Quality 

Methodological quality assessment of the retrieved SRs was performed using a modified version of the Assessing the Methodological Quality of Systematic reviews II tool (AMSTAR 2) [37]. This version comprises 13 items evaluating the quality of the retrieved literature, the risk of bias assessment, the quality of statistical analyses and reporting of results and transparency of potential sources of conflict of each study. Studies were rated on a scale from high quality to very low quality, based on the existence of critical and non-critical methodological weaknesses. 

### 2.5. Summarisation of Data Results and Conclusion Drawing

Based on the systematic description and on the quality assessment of the selected reviews included in synoptical tables (see Appendix A), at least two authors summarised the results and drew a conclusion for each disease.

## 3. Results

A total of 349 articles were identified for the seven diseases (Figure 1). After removal of 14 duplicates, 335 articles were screened by title and abstract. Of the 335 articles, 203 did not match our inclusion criteria and were thus excluded. For assessment 132 full-text articles were obtained and finally 73 SRs (with or without MAs) of prospective cohort studies and RCTs were included in qualitative synthesis (see Appendix A). Main reasons for the exclusion of 59 assessed full-text articles were an irrelevant study type (e.g., umbrella/narrative review, only/mainly cross-sectional and/or case-control studies) or an irrelevant outcome (e.g., prevention of falls in elderly, quality of life, poststroke depression). More detailed reasons for exclusion are shown in the Appendix A.

### 3.1. Respiratory Tract Diseases

#### 3.1.1. Asthma


**Background**


Asthma is a chronic inflammatory disorder of the airways that is characterised by recurring symptoms such as shortness of breath, wheeze, chest tightness, cough (varying over time and in intensity), as well as variable expiratory airflow limitation and often attended by increased serum immunoglobulin E (IgE) levels. A variety of factors may cause asthma exacerbations, including allergens, pollutants, and poor adherence to prescribed medications, while viral upper respiratory infection is the most predominant reason [38,39]. Virus-induced asthma exacerbations are associated with increased production of pro-inflammatory cytokines such as interleukin (IL)-17, which exacerbate allergic airway responses [40]. These cytokines are also associated with the severity of asthma and steroid responsiveness [41]. Acute exacerbations are major causes of morbidity and mortality [42,43]. 

Asthma affects more than 300 million people of all ages and all ethnic backgrounds, and it is estimated to cause 400,000 deaths annually [39,42,44]. Over the past few decades, the prevalence of asthma has increased steadily worldwide, especially in children and young adults of high-income countries [45,46,47]. Asthma is the most common chronic disease of childhood, affecting approximately 10% of children, with prevalence varying by definition and country of origin [48]. Until now, no convincing preventive strategies have been identified, and evidence concerning modifiable risk factors is inconsistent [49]. To improve clinical outcomes, it is important to prevent asthma exacerbations [38]. 

Vitamin D deficiency has been associated with increased incidence [50] and severity [51] of childhood asthma. Regarding the potential role of vitamin D and asthma, it should be considered that some data suggest an association between passive smoking and vitamin D deficiency in children with asthma [52]. Vitamin D has immunoregulatory properties [53,54] and plays an important role in inflammation [55]. Airway epithelial cells express the activating enzyme 1α-hydroxylase, which catalyses the formation of calcitriol from 25(OH)D [56,57,58]. VDRs, for which calcitriol is a known ligand, are present in cells of the immune system such as macrophages, dendritic cells and activated T- and B-cells [59]. Calcitriol shifts the balance of T lymphocyte response from T helper (Th)1 phenotype to Th2 phenotype [60,61]. In addition, calcitriol can suppress the proinflammatory cytokine IL-17 and the IL-4-mediated expression of IL-13 [62,63]. Calcitriol may also act directly on CD4+ T cells to promote T-regulatory cells (Tregs) that secrete the anti-inflammatory cytokine IL-10 [62,63,64,65,66]. Here, we synthesise results of SRs regarding vitamin D and asthma outcomes. 


**Results**


In our search for SRs on vitamin D and asthma risk, we identified 18 articles [38,67,68,69,70,71,72,73,74,75,76,77,78,79,80,81,82,83] of which 10 reported data on cohort studies [67,68,69,70,71,72,73,74,75,76] and eight reported data on RCTs [38,77,78,79,80,81,82,83] (Appendix A). Out of the 18 SRs, 14 were restricted to children [67,69,70,71,72,73,74,75,76,77,78,79,80,81], three included both, studies in children and adults [38,68,82], and one was based on individual participant data (IPD) with individuals aged 1.6–85.0 years [83]. Overall, the majority of the SRs were of high quality, assessed by AMSTAR 2 tool. We also identified a large Mendelian randomisation study on 146,761 study participants [35].


**Primary Prevention**


The aforementioned 10 SRs on cohort studies [67,68,69,70,71,72,73,74,75,76] were all primary prevention studies, performed in early childhood. Three out of the 10 SRs did not perform MAs [67,72,76]. Of the remaining seven MAs, six compared high versus low 25(OH)D levels [68,69,70,71,74,75]. No significant association between vitamin D status and asthma risk was reported in four of the six studies [70,71,74,75], whereas two studies [68,69] reported a significantly lower risk at higher 25(OH)D levels. Another SR [73] analysed nonlinear dose–response relationship by restricted cubic spline model. Data indicate a significant U-shaped association between maternal 25(OH)D levels and offspring asthma risk with the lowest risk at approximately 60–70 nmol/l (24–28 ng/mL) of 25(OH)D. Regarding study quality, seven were of high, one of moderate, and two of low or very low quality (Appendix A).

One out of the eight two-step MAs on RCTs focused on primary prevention [81] and used as an unspecific surrogate of asthma risk data on recurrent wheezing in offspring whose mothers participated in vitamin D supplementation studies. It was suggested that vitamin D may have a beneficial effect on recurrent asthma in children. This MA included three RCTs that used daily vitamin D_3_ doses of 60 µg and 100 µg (2400 and 4000 IU), respectively, or a single vitamin D_3_ bolus of 5000 µg (200,000 IU). The MA was of moderate quality (Appendix A). 

The aforementioned large Mendelian randomisation study [35] investigated four single nucleotide polymorphisms (SNPs) influencing 25(OH)D transport (GC globulin (vitamin D-binding protein)), synthesis (7-dehydrocholesterol reductase), hepatic hydroxylation (CYP2R1), and catabolism (CYP24A1), to estimate the association of genetically determined 25(OH)D and the risk of asthma. None of the four 25(OH)D-lowering alleles was significantly associated with asthma or elevated IgE levels, and there was also no significant association between genetically determined 25(OH)D and risk of asthma or IgE levels. 


**Therapeutic or Adjuvant Vitamin D Supplementation in Asthma Patients**


Six SRs performed a two-step MA on vitamin D supplementation in asthma patients [38,77,78,79,80,82]. Four out of these six MAs reported beneficial [77,78,79,82] and two non-significant effects [38,80]. Five MAs used asthma exacerbations as outcome [38,77,78,79,82], and one [80] relied on the lung function parameter FEV (forced expiratory volume). None of the six MAs performed subgroup analysis according to baseline 25(OH)D levels. Study quality was considered as moderate in two SRs and high in four SRs (Appendix A). In total, the six MAs were based on 15 RCTs. The vitamin D_3_ doses in the included RCTs were very heterogeneous and ranged between daily doses of 10 µg to 100 µg (400–4000 IU) or bolus doses of 1500 µg to 3000 µg (60,000–120,000 IU).

The identified IPD MA [83] was based on seven RCTs (five RCTs enrolled children, two RCTs enrolled adults), including 978 asthma patients, of whom 955 had outcome data (Appendix A). This analysis showed that vitamin D_3_ supplementation reduced the adjusted incidence rate ratio (aIRR) of asthma exacerbations requiring treatment with corticosteroids to 0.74 (95% CI: 0.56–0.97; *p* = 0.03) and the results were considered as high-quality evidence. Subgroup analyses suggested the possibility that protective vitamin D effects may only be seen in participants with baseline 25(OH)D of less than 25 nmol/l (10 ng/mL) (aIRR = 0.33; 95% CI: 0.11–0.98; *p* = 0.046) and not in participants with higher baseline 25(OH)D levels (aIRR: 0.77; 95% CI: 0.58–1.03; *p* = 0.08). However, the *p*-value for interaction of 0.25 did not provide definitive evidence for an interaction of baseline 25(OH)D with the vitamin D effect on asthma exacerbations. Study quality could not be assessed, because the AMSTAR 2 tool was not developed for use in IPD MAs [37].


**Discussion and Conclusion**


The majority of cohort studies were primary prevention studies in early childhood, whereas the majority of RCTs were performed in patients with already existing asthma. Results of MAs on the association between asthma risk and vitamin D status are inconclusive. This may be due to a potential U-shaped relationship between 25(OH)D and asthma risk [73], which remains uncovered in MAs that compared high versus low 25(OH)D levels to estimate the vitamin D effect on asthma risk [68,69,70,71,72,74,75]. The only MA of RCTs on primary prevention [81] supports the assumption that vitamin D may reduce asthma risk in early childhood, but this MA is limited by the use of an unspecific surrogate endpoint (wheezing). A Mendelian randomisation study [35], however, is not in line with a beneficial vitamin D effect on asthma risk. Nevertheless, it is noteworthy that the four SNPs could only explain 0.13%, 0.12%, 0.09%, and 0.02% of the variance in 25(OH)D [35] and are therefore subject to weak instrument bias [84]. Thus, the null effect reported in that study does not necessarily exclude a causal association between vitamin D status and asthma risk.

Regarding RCTs in asthma patients, the majority of MAs support the assumption of a beneficial vitamin D effect. Importantly, subgroup analyses of an IPD MA raise the hypothesis that protective vitamin D effects might be restricted to individuals with baseline 25(OH)D of less than 25 nmol/l (10 ng/mL). Besides heterogeneity in vitamin D dosing, baseline 25(OH)D level is an issue that needs consideration and long-term future RCTs should only be performed in populations with a high prevalence of vitamin D deficiency to draw conclusions concerning the impact of vitamin D supplementation for asthma risk. Another limitation is that most results are based on MAs in children. Therefore, caution is necessary when translating the results to the adult population. Two RCTs (not yet published) regarding the prevention of asthma exacerbations in children were identified at clinicaltrials.gov (identifier: NCT02687815 and NCT03365687) and may provide additional information on the effect of vitamin D in asthma patients. Overall, the majority of published SRs were of high quality.

In conclusion, adequate vitamin D status in childhood may reduce the risk of asthma exacerbations. Regarding vitamin D and asthma in the adult population, available data are insufficient to draw reliable conclusions.

#### 3.1.2. Chronic Obstructive Pulmonary Disease


**Background**


Chronic obstructive pulmonary disease (COPD) is a progressive, systematic inflammatory illness characterised by chronic airflow limitation. COPD patients suffer from reduction of lung function, loss of exercise capacity, frequent disease exacerbations, and development of extra-pulmonary comorbidities—such as osteoporosis, infection, and cardiovascular disease [85]. COPD exacerbations are commonly triggered by respiratory viruses and bacteria, which increase airway inflammation [86]. Episodes of acute worsening of symptoms are associated with increased mortality [87]. 

COPD affects more than 170 million people worldwide and caused an estimated 3.2 million deaths in 2015 [88]. COPD is the fourth leading cause of mortality globally [89] and is expected to become the third leading cause of death by 2020 [87]. Tobacco smoking is considered to be a major risk factor of COPD [90], but only 10–15% of long-term smokers develop symptomatic airflow obstruction [91].

Besides genetic susceptibility for COPD, poor vitamin D status has also been discussed as playing a role in the development of the disease [92]. Vitamin D has immunomodulatory properties [93], and calcitriol plays a role in modulating functions of the innate and adaptive immune systems [94]. Briefly, vitamin D signalling may stimulate innate immunity by upregulating antimicrobial peptide production and may suppress adaptive immunity by decreasing pro-inflammatory and increasing anti-inflammatory cytokine expression [59,95]. Cohort studies reported a positive association between circulating 25(OH)D and pulmonary function [96,97]. However, when interpreting observational studies on vitamin D and COPD, it should be considered that some investigations indicate that smoking is associated with lower 25(OH)D concentrations [98]. Nevertheless, additional evidence for a potential link between vitamin D and COPD is provided by a MA of RCTs on vitamin D supplementation and acute respiratory tract infections [99]: vitamin D supplements reduced infections significantly and subgroup analysis indicated that this effect may be highest in individuals with deficient vitamin D status at study inclusion, i.e., circulating 25(OH)D concentrations < 25 nmol/l (10 ng/mL). Thus, vitamin D may indirectly influence COPD exacerbations by reducing the risk of airway infections. 

Results of SRs regarding vitamin D and COPD outcomes are summarised below.


**Results**


In our search for SRs on vitamin D and COPD risk, we identified six SRs [31,100,101,102,103,104] (Appendix A) of which three reported MAs on pooled data regarding vitamin D status or vitamin D supplementation and COPD risk [101,102,104]. The study quality assessed by the AMSTAR 2 tool was very heterogeneous (Appendix A). No Mendelian randomisation studies were identified.


**Primary Prevention**


Our literature search did not identify SRs of prospective cohort studies or RCTs on the primary prevention of COPD by vitamin D. 


**Therapeutic or Adjuvant Vitamin D Supplementation in COPD Patients**


Of the three SRs reporting MAs, the analysis by Zhu et al. [101] included 18 studies, of which five were case-control studies, eight were cohort studies, and five were RCTs. Both, the case-control and cohort studies compared 25(OH)D concentrations between COPD patients and controls. Since these studies used a cross-sectional approach, they are of limited relevance for our review. The five RCTs of that MA were not evaluated by the authors using the statistical approach of a MA, but by narrative description of study results only. They concluded that four RCTs showed beneficial vitamin D effects in COPD patients, at least in those with circulating 25(OH)D concentrations less than 50 nmol/l (20 ng/mL). Another MA by Zhu et al. [102] included 21 observational studies. The vast majority of studies were case-control studies that compared vitamin D status of COPD patients with controls or vitamin D status with COPD risk/severity. Zhu et al. performed a MA of seven studies (two cohort studies, five case-control studies) showing that patients with severe to very severe COPD had lower serum 25(OH)D levels compared with patients with mild to moderate COPD (SMD: −0.87, 95% CI: −1.51, −0.22). A MA of five studies (two cohort studies, three case-control studies) indicated that acute exacerbation COPD (AECOPD) patients had lower levels of serum 25(OH)D compared to stable COPD patients (SMD: −0.43, 95% CI: −0.70, −0.15). Three studies (two case-control studies, one cohort study; thus, failing to meet our inclusion criteria) were included in an evaluation of the association between vitamin D deficiency and COPD exacerbations, revealing no significant association (odds ratio: 1.17, 95% CI: 0.86–1.59) [102]. However, since these studies included by Zhu et al. mostly used a cross-sectional approach, they are of limited relevance to our review. Of the three SRs not reporting MAs [31,100,103], the SR by Autier et al. [100], which was based on two prospective studies, stated an inverse association between circulating 25(OH)D and COPD exacerbations. On the contrary, in a SR by Ferrari et al. [103], based on five prospective cohort studies, no association could be shown between exacerbations frequency and circulating 25(OH)D. Nevertheless, from the RCTs included in that SR, it was concluded that especially a group of patients with low 25(OH)D level, i.e., < 50 nmol/l (20 ng/mL), may benefit from vitamin D supplementation. Similarly, another SR [31] reported significant reductions in pulmonary exacerbations of COPD patients by vitamin D supplementation only at low 25(OH)D concentrations. 

An IPD MA by Jolliffe et al. [104] identified four RCTs, of which three (472 randomised patients) were included in their analysis. The MA concluded that vitamin D supplementation did not influence overall rate of moderate/severe COPD exacerbations (adjusted incidence rate ratio 0.94 (95% CI: 0.78 to 1.13)). However, subgroup analysis indicated protective vitamin D effects in patients with baseline 25(OH)D concentrations < 25 nmol/l (10 ng/mL) (adjusted incidence rate ratio 0.55 (95% CI: 0.36 to 0.84)).

The study quality assessed by the AMSTAR 2 tool was very heterogeneous for the included SRs and MAs (Appendix A). One SR [104] was an IPD MA, for which AMSTAR 2 was not developed. Overall, RCTs were included in four SRs. All RCTs included in the SRs [105,106,107,108,109] investigated effects of vitamin D supplementation as primary exposure. Study duration was six weeks in one study [105], six months in two studies [106,107], and one year in two RCTs [108,109]. In four RCTs [105,107,108,109], the vitamin supplement was D_3_ and in one study it was not specified whether vitamin D_2_ or D_3_ was used [106]. Daily and bolus vitamin D administration was performed in two [105,107] and three RCTs [106,108,109], respectively. The vitamin D dose ranged between 20 µg per day (800 IU) [105] and 2500 µg (100,000 IU) per 4 weeks [106,108]. The calculated mean daily dose ranged between 20 µg (800 IU) [105] and 89 µg (3560 IU) [106,108]. Mean baseline 25(OH)D concentrations were reported in four RCTs [105,107,108,109] and were below 50 nmol/l (20 ng/mL) in all these four studies. In-study 25(OH)D concentrations were also presented in these four RCTS, of which three RCTs exceeded a mean in-study 25(OH)D concentration of 75 nmol/l (30 ng/mL) [107,108,109], whereas the in-study 25(OH)D concentration remained on average below 75 nmol/l (30 ng/mL) in the 6-week study with a daily dose of 20 µg (800 IU) vitamin D [105].


**Discussion and Conclusion**


Data on primary prevention of COPD by vitamin D are scarce. Results of SRs on prospective cohort studies regarding the association of vitamin D status on COPD exacerbations are inconclusive and no reliable MAs are currently available on this topic. With respect to RCTs in COPD patients, only the SR by Jolliffe et al. [104] analysed the data according to the approach of a MA. Results indicate a potentially beneficial vitamin D effect on COPD exacerbations solely in patients with deficient 25(OH)D concentrations—i.e., concentrations below 25 nmol/l (10 ng/mL)—but not in patients with higher baseline 25(OH)D concentrations. Although results are based on a pre-specified subgroup analysis [104], caution is necessary in interpreting these results. The medical literature is replete with exciting secondary end points that have failed when they were subsequently formally tested as primary end points in adequately powered RCTs [110]. Moreover, two out of the three studies included in this IPD MA [108,109] used high dose bolus administration of vitamin D. Formally, these studies have to be considered as phase 2 clinical trials, which are usually performed for drug approval. Nevertheless, it is also noteworthy that the recommended daily intake for adequacy by nutrition societies of 10 µg to 20 µg (400–800 IU) vitamin D (by supplements) [2,3,4,5] are able to increase circulating 25(OH)D concentrations of 25 nmol/l to 50 nmol/l (10–20 ng/mL) [111] and thus into the range, which many nutrition societies consider adequate [2,3,4,5]. 

In conclusion, the quality of SRs on COPD patients is very heterogeneous. At present, the effect of vitamin D on the primary prevention of COPD is unclear. Based on a relatively small number of RCTs and participants, vitamin D-deficient patients with already existing COPD probably benefit from vitamin D supplementation. As identified at clinicaltrials.gov, two RCTs on the effects of vitamin D supplementation in COPD patients are expected to report additional data on this topic (identifier: NCT02122627 and NCT03781895). 

#### 3.1.3. Acute Respiratory Tract Infections


**Background**


Acute respiratory tract infections (ARI) comprise a group of infections that can occur in the upper and lower respiratory tract and are leading causes of global morbidity and mortality in children and adults [112]. The upper respiratory tract infection is one of the most common acute illnesses in the outpatient setting and is often characterised by a mild and self-limited disease course. Symptoms associated with upper respiratory tract infections are irritations and swelling of the nose, sinuses, pharynx, larynx, and the large airways. In contrast, lower respiratory tract infections affect the bronchial tubes and the lungs and can cause bronchitis, pneumonia, and pulmonary tuberculosis.

Lower respiratory tract infections are the fifth leading cause of death and the leading infectious cause of death worldwide [113]. Lower respiratory tract infections mainly affect children and individuals older than 65 years. In 2015, pneumonia accounted for 15% of the deaths in children under 5 years worldwide [114]. Approximately 45% of all community-acquired pneumonia occur in patients aged >65 years [115,116,117]. Pneumonia is an inflammation of the lung that compromises gas exchange in the lungs and leads to symptoms such as cough, fever, and breathing difficulties [118]. ARIs are the most common reason for antibiotic therapy in adults [119].

The most common etiological factors that are responsible for ARIs are bacteria and viruses. Risk factors of pneumonia are crowded living conditions, malnutrition, HIV infection, lack of breastfeeding in infants, lack of immunisation, chronic health conditions, and exposure to tobacco smoke or indoor air pollutants [112]. Vitamin D exerts multiple effects in the immune system which includes the synthesis of antimicrobial peptides and modulation of T cell system [120,121]; more importantly, activation of toll-like receptors upregulates the VDR and vitamin D hydroxylases in human macrophages and the production of the antimicrobial cathelicidin which can kill intracellular *Mycobacterium tuberculosis* [122]. These data support a link between vitamin D and the innate immune system and justify the assumption that vitamin D can impact the prevalence and course of ARIs. Observational studies have reported an association between vitamin D status and ARIs [123,124,125]. However, the benefits of vitamin D for prevention and treatment of ARI are ambiguous. Thus, we aimed to summarise data of SRs regarding vitamin D and ARI outcomes.


**Results**


In our search for SRs on vitamin D and ARI, we identified 14 records [71,74,77,100,126,127,128,129,130,131,132,133,134,135]; of which five reported data on cohort studies; [71,74,100,126,128] and 11 included MAs from RCTs [77,100,126,127,129,130,131,132,133,134,135] (Appendix A). Using AMSTAR 2 as an instrument for assessing methodological quality, 8 of the 14 SRs that examined the association between vitamin D and the prevention or treatment of ARI can be categorised as high quality; three studies were categorised as medium quality; and the other two studies as low and very low quality, whereas AMSTAR 2 was not developed to assess the quality of the only IPD MA. We could not identify Mendelian randomisation studies on vitamin D and ARI.


**Primary Prevention**


Regarding the association of vitamin D status and risk of ARI, we identified five SRs of cohort studies with very heterogeneous study quality according to AMSTAR 2 assessment [71,74,100,126,128]. In adults, Autier et al. found two studies reporting on an inverse association between serum 25(OH)D and respiratory infections and one study showing an inverse association between serum 25(OH)D and days of absence due to respiratory infections [100]. Similar results in adults were reported by Jolliffe et al. [126]. Some SRs on prenatal vitamin D status (assessed by maternal or cord blood 25(OH)D mainly derived from birth cohort studies) and ARI reported mixed results [71,74,126,128]. In the MA by Pacheco-Gonzalez et al., the pooled odds ratio for ARI in the offspring was 0.64 (95% CI 0.47 to 0.87) when comparing the highest with the lowest 25(OH)D category [74], whereas there was no significant result for such an association in the MA by Feng et al. [71]. 

Regarding data on RCTs reporting effects of vitamin D supplementation on risk of ARI, we identified nine SRs with heterogeneous, yet largely high study quality [77,100,126,130,131,132,133,134,135]. Among these studies, the largest MA, and the only one following an IPD approach, was published by Martineau et al. [99,135]. They included 25 RCTs with a total of 11,321 participants aged 0 to 95 years including 10,933 participants with available IPD data. The main outcome was that vitamin D supplementation reduced the risk of at least one ARI with an adjusted odds ratio of 0.88 (95% CI: 0.81 to 0.96; *p* = 0.003). In subgroup analyses, it was shown that this protective effect was only significant in those individuals receiving daily (7.5 to 100 µg (300–4000 IU)) or weekly vitamin D doses (35 to 500 µg (1400–20,000 IU)), but not in those receiving any bolus doses (i.e., one bolus of at least 750 µg (30,000 IU) vitamin D; range 750 to 10,000 µg (30,000–400,000 IU)). Moreover, the protective effect was stronger in individuals with baseline 25(OH)D below 25 nmol/l (10 ng/mL) as compared to those with 25(OH)D concentrations ≥25 nmol/l (10 ng/mL), whereas no other effect modifiers were identified. Regarding the clinical effect size, the overall number needed to treat was 33 (95% CI: 20 to 101) and dropped to 8 (95% CI: 5 to 21) in individuals with baseline 25(OH)D concentrations below 25 nmol/l (10 ng/mL). Martineau et al. concluded that the evidence contributing to the findings of their MA was of high quality. It should, however, be stressed that there was a very high heterogeneity across the studies and funnel plot analysis showed a degree of asymmetry suggesting that small RCTs showing adverse effects might not have been included or published. Findings from the other MAs that included significantly fewer participants compared to the work by Martineau et al. were inconsistent, showing either a protective effect of vitamin D supplementation [126,130,131] or no effect [77,100,126,132,133,134]. Importantly, the vast majority of the included original studies of these MAs were also included in the work by Martineau et al. [99,135].


**Therapeutic or Adjuvant Vitamin D Supplementation in Patients Suffering from ARI**


Regarding data on RCTs reporting effects of vitamin D supplementation on the outcome of pneumonia in children, we identified two SRs with moderate to high study quality [127,129]. In the only MA on this topic, there was no significant effect of vitamin D_3_ supplementation, neither on time to resolution of acute pneumonia nor on duration of hospitalisation [129]. Thus, data from RCTs on vitamin D supplementation and ARI document no significant safety concerns regarding vitamin D.


**Discussion and Conclusion**


Data from cohort studies found an inverse association between serum 25(OH)D and respiratory infections in adults. The main finding of our literature search is that the largest and well conducted IPD MA of RCTs reported a significant role of vitamin D supplementation to reduce the risk of ARI [135]. In subgroup analyses, this significant effect was, however, restricted to individuals who did not receive bolus doses of vitamin D and was particularly strong in individuals with 25(OH)D concentrations below 25 nmol/l (10 ng/mL). Effect modification by baseline 25(OH)D is biologically sound and consistent with the concept that vitamin D supplementation is most effective in individuals with severe vitamin D deficiency. Why bolus doses of vitamin D (in ranges between 750 to 10,000 µg (30,000–400,000 IU)) are ineffective for prevention of ARI remains speculative, but it has been hypothesised that wide fluctuations in vitamin D metabolites might cause adverse effects, including dysregulation of vitamin D metabolising enzymes [99]. Funnel plot analyses suggest that RCTs showing adverse vitamin D effects may not have been included in the MA. Another limitation is that the definitions of ARI were heterogeneous and hardly supported by virological, microbiological, or radiological confirmation. Some caution is therefore warranted with the final claim on vitamin D and prevention of ARI. In general, the preventive effect of vitamin D on ARI is, however, biologically plausible because VDR activation exerts numerous immunological effects that may translate to protection against ARI. Furthermore, epidemiological studies have also largely—although not consistently—supported the concept that vitamin D may protect against ARI. Camargo et al. recently (after the search period) published data from a pre-specified analysis of the ViDA (Vitamin D Assessment) study that aimed to investigate the effects of monthly high-dose vitamin D_3_ supplementation (initial oral dose of 5000 µg (200,000 IU) vitamin D_3_ followed by 2500 µg (100,000 IU) monthly) on ARI prevention in more than 5000 older adults [136]. The results of this RCT failed to show a preventive effect of vitamin D supplementation on ARI. However, it should be noted that these findings were obtained from a study collective with a low prevalence of vitamin D deficiency.

Regarding treatment of ARI, the current literature including RCTs in children indicates that there is no significant effect of vitamin D. 

Several RCTs on the effects of vitamin D supplementation on the prevention and treatment of ARI are expected to report additional data on this topic in the future (e.g., clinicaltrials.gov identifier NCT02185196; NCT02054182; NCT02046577; NCT03799406). However, most of these RCTs do not exclusively target vitamin D deficient individuals and some use bolus doses. Therefore, these RCTs have a high probability of documenting neutral effects of vitamin D supplementation on ARI in view of effect modification by baseline 25(OH)D and the use of bolus doses as reported by Martineau et al. [99,135]. 

The majority of SRs that examined the association between vitamin D and the prevention or treatment of ARI had a moderate or high methodological quality. Current evidence from MAs of RCTs suggests that vitamin D supplementation may prevent ARI. This effect was only evident in individuals who did not receive bolus doses of vitamin D and was particularly strong in those with baseline serum 25(OH)D concentrations below 25 nmol/l (10 ng/mL). Complementary to this finding, SRs of cohort studies indicated an inverse association between vitamin D status and ARI. Current data from RCTs suggest that vitamin D supplementation has no significant effect on the treatment of ARI. 

### 3.2. Neurodegenerative and Mental Diseases

#### 3.2.1. Dementia and Cognitive Decline


**Background**


Dementia is a clinical syndrome characterised by global cognitive impairment with a decline in memory and at least one other cognitive domain, such as language, visuospatial, or executive function. It is a chronic, malignant, and continuously progressing disease, associated with impairment in functional abilities and in many cases behavioural and psychiatric disorders, leading invariably to dependence on others [137]. Development of dementia from normal cognition is a continuous, slow process of cognitive decline over many years which is difficult to distinguish from normal ageing. Diagnosis of dementia is based on comprehensive assessment of symptoms which need to be persistent for at least six months. The two most common causes of dementia are Alzheimer’s disease and cerebrovascular disorders, which often overlap. Worldwide, currently around 50 million people are affected with nearly 10 million new cases expected every year [138]. The prevalence of dementia is mounting with increasing age. In Germany, only 1.5% of currently 1.7 million people with dementia are younger than 65 years; 16% of those aged 80–84 years and more than 40% of those aged 90 years or older are affected [139]. Since no effective treatment is available at present, maintenance of cognitive abilities into old age and the prevention of cognitive decline are a major public health concern.

In recent years, the association between vitamin D and dementia or neurocognitive decline has attracted growing interest, and evidence is accumulating for potential neuroprotective effects of vitamin D. Data from cross-sectional analyses suggest that low serum concentrations of 25(OH)D are associated with increased risk of Alzheimer’s disease and other forms of dementia and cognitive impairment [140,141]. In addition, there is evidence from animal models and in vitro studies that vitamin D may influence the development of neurodegenerative disorders. Exact mechanisms are unclear, but evidence suggests that it may protect against cognitive dysfunction through its effect on synaptic plasticity, immune modulation, neuronal calcium regulation, and enhanced nerve conduction [141,142,143]. In addition, vitamin D may affect vascular brain disease by mediating harmful effects of inflammation, calcium dysregulation and increased oxidative stress and also by modulation of vascular disease risk factors such as elevated blood pressure [141]. Moreover, some data suggest that vitamin D might exert some beneficial effects with relevance for cardiometabolic health, including antiatherogenic effects, improvement of endothelial function, arterial elasticity and metabolic profile and inhibition of the renin-angiotensin-aldosterone system, which could be another mechanism for the possible protective effects of vitamin D against dementia and cognitive decline [144].

The relationship between serum 25(OH)D status and dementia and/or cognition has been examined in many cross-sectional studies, in an increasing number of longitudinal studies and in a few interventional studies with conflicting results. Several SRs and MAs have been performed in the last 10 years. Results of SRs regarding vitamin D and dementia and cognitive decline outcomes are summarised below.


**Results**


13 SRs (including eight MAs) [100,145,146,147,148,149,150,151,152,153,154,155,156] on vitamin D and risk of dementia and/or cognitive decline were included in this analysis (Appendix A). In addition, four Mendelian randomisation studies were considered [157,158,159,160]. The quality of the publications according to AMSTAR 2 is summarised in Appendix A. All articles addressed preventive effects of vitamin D status or supplementation in adults without diagnosed dementia. No reviews or relevant studies were identified regarding therapeutic effects of vitamin D in patients with dementia. 


**Results from SRs of Cohort Studies**


All identified 13 SRs examined cohort studies, six with the outcome dementia [148,149,150,152,153,155] (including three with the outcome Alzheimer’s disease) [148,152,153] and eight with the outcomes cognition, cognitive impairment or cognitive decline [100,145,146,147,149,151,154,156] using single or combinations of several neuropsychological tests. 

The most recent and also largest high-quality MA regarding dementia risk included data from 10 cohort studies with 28,640 participants (mean/median age of 56 to 85 years) and follow-up periods of 2–21 years [153]. A significant inverse association was found between 25(OH)D concentrations and the risk of dementia (RR 0.72, 95% CI: 0.59–0.88; I^2^ = 33%, comparison highest vs. lowest 25(OH)D categories) and the risk of Alzheimer’s disease (RR 0.78, 95% CI: 0.60–1.00; I^2^ = 57%). In addition, a dose-response analysis revealed that the risk of dementia decreased by 5% (RR 0.95, 95% CI: 0.93–0.98) and the risk of Alzheimer’s disease by 7% (RR 0.93, 95% CI: 0.89–0.97) for every 10 nmol/l (2.5 ng/mL) increase in 25(OH)D concentrations [153]. Jayedi et al. [152] meta-analysed eight of these cohort studies and reported an increased risk of dementia in vitamin D-deficient (defined as <25 nmol/l (10 ng/mL); n = 5; HR 1.33) but not in vitamin D-insufficient participants (25–50 nmol/l (10–20 ng/mL); n = 6). In addition, a 17% reduced risk of dementia (n = 7) and of Alzheimer’s disease (n = 6) per 25 nmol/l (10 ng/mL) increase in serum 25(OH)D concentrations was found, however with a large heterogeneity between studies (I^2^ = 81 and 82%, respectively). Another high-quality MA of five of these cohort studies reported a 1.54 times increased risk of dementia in adults with serious deficiency (<25 nmol/l (10 ng/mL)) compared to sufficient supply (≥50 nmol/l (20 ng/mL)) [150]. Two MAs and one SR published in 2015 and 2016 of low or very low quality with only two or three cohort studies, respectively, also reported an increased risk of dementia in adults with low 25(OH)D concentrations at baseline [148,149,155].

The most recent and largest high-quality MA addressing cognition in middle-aged and older adults without a diagnosis of dementia included 14 prospective cohort studies [151]. The probability of cognitive decline was higher in participants with low 25(OH)D concentrations than in those having higher 25(OH)D levels (OR 1.26; CI 1.09–1.23). Heterogeneity between the study effect sizes was again large (I^2^ = 75%) and the possibility of publication bias was rated as high [151]. Other available MAs are older (publication dates 2012 and 2013), of lower quality and included only three [147] and two [145] of these prospective cohort studies. They focused on executive functions and also found a higher risk of cognitive decline over four to seven years in older adults with low 25(OH)D concentrations at baseline (heterogeneity 0% and 10%) [145,147]. A SR (low quality) on the association between vitamin D status and cognition including six prospective cohort studies reported a significant decline in one or more cognitive function tests in participants aged 65 years or older with lower 25(OH)D concentrations compared to participants with higher 25(OH)D levels in four of these six studies [154]. The SR (low quality) of Autier et al. [100] included four cohort studies and concluded that frequency of cognitive decline is increased in participants with low 25(OH)D concentrations. Balion et al. [146] included only two cohort studies in their review (high quality) and reported conflicting results on cognitive outcomes. 


**Results from SRs of RCTs**


Among the SRs with focus on cognition, six [100,145,146,147,151,156] (two high, one medium, three low quality) included up to three intervention studies in their analyses. In total, nine studies were evaluated, reporting conflicting results: seven RCTs (two using multi-nutrient supplements) and two pre-post studies in very heterogeneous populations (adolescents, young adults, older ambulatory people with history of falling, healthy older people, nursing home residents). Supplementation modes (e.g., daily, weekly or monthly oral supplementation, single injection, vitamin D_2_ or vitamin D_3_, and vitamin D_3_ plus calcium), and dosages differed widely between the studies. The latter ranged from 10 µg per day (400 IU) vitamin D_3_ [100] to 125 µg per day (5000 IU) vitamin D_3_ and to a bolus dose of 15,000 µg (600,000 IU) vitamin D_2_ [147,151]. In addition, sample sizes were mostly small and supplementation periods were short. Only one larger placebo-controlled intervention study is described in one SR [100], which investigated the effects of 10 µg per day (400 IU) of vitamin D_3_ and 1000 mg per day of calcium carbonate supplementation on cognitive outcomes in 4143 women aged 65 and older without probable dementia at baseline. Mean follow-up was 7.8 years. There were no significant differences in incident dementia or mild cognitive impairment or in global or domain-specific cognitive function between verum and placebo groups [161]. 

Two of the abovementioned reviews performed a MA. Goodwill et al. [151] (high quality) included two studies (I² = 35%) and Annweiler et al. [147] (moderate quality) three intervention studies (I² = 49%). Both analyses found no effect of vitamin D supplementation on cognitive parameters. 


**Results from Mendelian Randomisation Studies**


We identified four publications that applied a Mendelian randomisation approach to address causal inference [157,158,159,160]. Kuzma et al. [160] investigated the association between any risk factor and global cognitive function, all-cause dementia or dementia subtypes. Genetic evidence supported a causal association between telomere length and Alzheimer’s disease, whereas limited evidence for other risk factors including vitamin D status was found. Mokry et al. [158] provided evidence that genetically decreased 25(OH)D concentrations are associated with increased risk of Alzheimer’s disease. Here four SNPs were analysed, that combined described 2.44% of the variance in circulating 25(OH)D levels in the SUNLIGHT (Study of Underlying Genetic Determinants of Vitamin D and Highly Related Traits) study [158]. Alfred et al. [157] investigated whether genetic variants influencing 25(OH)D concentrations are associated with cognitive capability in middle-aged and older adults. They observed a negative association between the allele of rs2282679 (GC globulin), which is associated with higher 25(OH)D concentrations, and word recall. A Mendelian randomisation study of Maddock et al. [159] investigated the causal nature of the association between serum 25(OH)D concentrations and cognitive function in mid- to later life but found no evidence for such an association.


**Discussion and Conclusion**


In summary, results from SRs of cohort studies report an increased risk of dementia including Alzheimer’s disease and for cognitive decline in mainly older persons (>65 years) with low 25(OH)D concentrations; however, with a large heterogeneity in study effect sizes. In addition, some of the Mendelian randomisation studies suggest a relationship between 25(OH)D concentration and the risk of Alzheimer’s disease and cognitive function. Only few very heterogeneous RCTs are available, which have not convincingly demonstrated a positive effect of vitamin D supplementation on cognitive outcomes. Except for one, these studies are small and of short duration; the only larger RCT found no difference between the verum and placebo group. 

Potential explanations for the differences between evidence from epidemiological observational studies, Mendelian randomisation studies and intervention studies may result from using varying methods (e.g., analytical and statistical approaches, vitamin D reference ranges), different baseline concentrations of 25(OH)D, as well as cognitive function, or the possibility that low 25(OH)D concentration in persons with dementia or cognitive impairment is only an epiphenomenon. In addition, the intervention studies conducted so far are very heterogeneous regarding study endpoints (e.g., cognitive function, cognitive impairment, dementia, Alzheimer’s disease) and participants (e.g., adolescents, healthy older adults, nursing home residents), are mostly of short-term duration and poor quality. Furthermore, a variety of neuropsychological tests were used for different aspects of cognitive function (e.g., general cognition, reasoning and language, figural creations, visuospatial abilities, mental speed/attention). 

Regarding dosage, a recent RCT [162] tested whether 50 µg per day (2000 IU) is more effective than 20 µg (800 IU) vitamin D_3_ for improving cognitive performance among relatively healthy adults aged ≥60 years (31% 25(OH)D < 50 nmol/l (20 ng/mL)), but found comparable results in both groups over a 24-month treatment period and no significant improvement in either group. Similarly, another recent RCT found that supplementing older adults with a history of falling (58% 25(OH)D < 50 nmol/l (20 ng/mL)) with 600 µg (24,000 IU) vitamin D_3_, 1500 µg (60,000 IU) vitamin D_3_ or a combination of 600 µg (24,000 IU) vitamin D_3_ with calcifediol once per month also led neither to different effects in mental health within one year, nor to any significant improvement [163]. Participants achieving the highest serum 25(OH)D levels (112.5–247.5 nmol/l (45–99 ng/mL)), however, had a “small, clinically uncertain but statistically significant improvement in mental health scores”, irrespective of the supplement dose. 

Unfortunately, many questions regarding the role of vitamin D in the development of dementia and cognitive decline are currently unanswered—e.g., what, if any, is the optimal blood concentration required to support neuroprotection? Is the association seen in observational studies mediated by other lifestyle factors, such as physical function or fitness? Thus, well-designed RCTs are required in different population groups (e.g., with adequate sample size and statistical power, vitamin D dosage and formulation, valid comprehensive cognitive assessments) to determine if and/or in whom vitamin D supplementation and increased 25(OH)D concentrations affect the varied aspects of cognitive health, and to what extent. This is particularly challenging with respect to dementia because of a very long prodromal stage and latency period. 

In conclusion, available SRs—as well as included primary studies—are of heterogeneous quality (Appendix A), and available scientific evidence from RCTs does not support a clear benefit of vitamin D supplementation in the prevention of dementia and cognitive decline. Nevertheless, based on the associations described in prospective cohort studies, it seems prudent to aim to prevent vitamin D deficiency in older adults as a viable component of brain health strategy. The results of several ongoing intervention studies, especially of the VIOLET-BUD Study (clinicaltrials.gov identifier: NCT03733418) may be helpful to better understand the role of vitamin D in the course of neurodegeneration. 

#### 3.2.2. Depression


**Background**


Depression is a leading cause of disability worldwide and an important contributor to the overall burden of disease. Worldwide more than 264 million people are affected [164]. Major depression is defined as a period of at least two weeks when a person experiences a depressed mood or loss of interest or pleasure in daily activities, plus several other symptoms, like problems with sleep, concentration, or self-worth. Dysthymia is a less severe, but persistent, form of depression. 

The biological plausibility for an association of vitamin D with depression comes from several observations. Within the brain, vitamin D is involved in regulating cytokines in the inflammation pathways and contributing to the release of neurotransmitters [150,165,166]. The VDR, which functions as a gene regulator, has been found in several brain regions implicated in depression [167,168,169,170]. In the hippocampus, an interaction between VDRs and glucocorticoid receptors, which are implicated in the stress response, has been described [171]. We therefore aimed to summarise results of SRs regarding vitamin D status or supplementation and prevention or therapy of depression.


**Results**


We identified 12 SRs [31,100,156,172,173,174,175,176,177,178,179,180] covering 31 original publications between 1998 and 2017, plus five SRs [181,182,183,184,185] on depression in women before and after childbirth (Appendix A). Most studies relied on dimensional self-ratings of depressive symptoms as outcome measures. The AMSTAR 2 study quality was moderate or high for 11 SRs, but low or very low for seven other SRs. Furthermore, one large Mendelian randomisation study was identified [186]. 


**Primary Prevention**


Two SRs performed a MA of observational cohort studies [172,173], four summarised data of RCTs [31,174,175,178], while two summarised the results of RCTs and cohort studies [100,156] and five SRs included a MA of RCTs [176,177,178,179,180]. Anglin et al. (2013) performed a moderate-quality MA of three cohort studies with a total of 8815 elderly subjects, followed up for 1–6 years, and found a two-fold increased risk of depression in those with the lowest compared to the highest vitamin D status [172]. Ju et al. included one additional large cohort study in their moderate quality SR, and found an inverse association between 25(OH)D concentrations and incident depression [173]. 

Some RCTs were done in specific groups at risk of depression (e.g., elderly subjects with established vitamin D deficiency, or with obesity). Shaffer et al. (2014) found no effect of vitamin D_3_ supplementation with 10 µg per day (400 IU) to 7500 µg once (300,000 IU) on the development of depressive symptoms in a high-quality MA of seven RCTs with 3191 at-risk subjects [177], and neither did Li et al. (2014) in their high-quality MA when analysing six RCTs in clinically depressed subjects [176]. Including these and three additional RCTs, comprising 4923 subjects with rather low depression levels at baseline, the high-quality MA of Gowda et al. (2015) also found no impact of vitamin D supplementation (10 µg per day—1250 µg per week (400 IU per day—50,000 IU per week)) on reducing depressive symptoms [179]. The authors of these SRs noted the significant heterogeneity of design and outcomes of these “preventive” RCTs. Spedding (2014) sorted 15 RCTs with 42,258 participants (mostly from at-risk groups) according to whether or not they had “biological flaws” (e.g., lack of 25(OH)D measurements, or lack of 25(OH)D change during vitamin D_3_ supplementation, or a sufficient 25(OH)D level in participants at baseline) [178]. RCTs without such validity issues, administering 25 to 375 µg (1000–15,000 IU) vitamin D_3_ per day, tended to show an improvement in depressive symptoms. However, this SR itself was judged to be of low quality applying the AMSTAR 2 criteria.

One large two-sample Mendelian randomisation study [186] examined whether common gene variants, together explaining 7.5% of the variance in serum 25(OH)D concentrations in the SUNLIGHT study [187], would be associated with a diagnosis of major depression (i.e., whether these SNPs would be less common in patients with this diagnosis than in psychiatrically healthy controls). Analysis of 59,851 cases with major depression and 113,154 controls from the Psychiatric Genetics Consortium database revealed no such association, suggesting that life-long (genetically determined) decreases in serum 25(OH)D concentrations do not increase the risk of major depression in generally healthy populations. However, the authors noted that their MR study had limited power to examine effects of extremes on either side of the serum 25(OH)D spectrum.


**Postpartum and Antepartum Depression (Primary Prevention)**


Pregnancy, the postpartum period, and in particular the perinatal period are associated with an increased risk of depression in women. In five SRs of mainly high quality, results have been summarised covering largely overlapping subsets of 11 original studies between 2010 and 2017 [181,182,183,184,185]. One SR summarised observational cohort studies [184], two SRs summarised the results of secondary analyses of a RCT and cohort studies [181,183], and one SR summarised the results of a RCT, secondary analyses of two RCTs and cohort studies [182]. In addition, one high-quality SR by Wang et al. (2018) included seven cohort studies, one case-control study and one RCT, which investigated the relationships between circulating levels or supplementation of vitamin D with antepartum or postpartum depression [185]. MAs were performed with data from the seven cohort studies. The first main MA including three studies which investigated the association between low 25(OH)D concentrations (<30 nmol/l (12 ng/mL)) and antepartum depression showed no significant association. However, the second main MA (including four studies), which studied the relationship between lower circulating levels of 25(OH)D (<50 nmol/l (20 ng/mL)) and postpartum depression, showed that lower circulating levels of 25(OH)D were associated with an increased odds ratio for postpartum depression (OR 3.67; 95% CI 1.72–7.85). In addition, relationships were observed in two subgroup analyses stratified by study region (Oceania: OR 2.00; 95% CI 1.31–3.06, Asia: OR 7.17; 95% CI 3.89–13.21). Evidence of statistical heterogeneity was found in all MAs. In addition, one RCT showed that participants given vitamin D supplements (50 µg (2000 IU)) in late pregnancy, as compared to the placebo group, had a greater reduction in depression scores at 38 to 40 weeks of pregnancy [185].


**Therapeutic or Adjuvant Vitamin D Supplementation in Depressed Patients**


In contrast to the largely negative intervention results in subjects with no or few depressive symptoms (see Primary Prevention above), a moderate-quality MA including four RCTs performed on 948 patients with major depression [180] found a consistent and medium-sized positive effect of vitamin D (oral 37.5 µg per day (1500 IU), or 1250 µg per week (50,000 IU), or up to 7500 µg (300,000 IU) single dose injection) versus placebo on depressive symptoms (SMD = 0.58, 95% CI: 0.45–0.72, *p* < 0.01, I^2^ = 0%). All these patient studies were conducted in Iran or China.


**Discussion and Conclusion**


SRs of longitudinal cohort studies suggest that low levels of 25(OH)D are associated with higher subsequent depressive symptoms in elderly subjects. A recently (after the search period) published dose–response MA [188] of six cohort studies (16,287 elderly subjects, 1157 cases with depression) confirmed this inverse association, and calculated that every 25 nmol/L (10 ng/mL) increase in serum 25(OH)D was associated with a 12% decrease in the risk of depression in older adults (I^2^ = 79.0%, *p* < 0.001). In addition, low circulating levels of 25(OH)D are associated with postpartum depression.

In contrast, Mendelian randomisation studies do not support a causal role of (genetically) lower levels of serum 25(OH)D on the development of depressive symptoms or major depressive disorders. Recently, two additional Mendelian randomisation studies concerning 25(OH)D and depression were published (after the search period) [189,190]. A Mendelian randomisation study analysing depressive symptoms and broadly defined depression in two large population studies (> 480,000 participants) did not find an association of the alleles linked to serum 25(OH)D with these depression phenotypes [189]. The Mendelian randomisation study with 339,256 UK biobank participants (including 23,294 cases with depression) by Meng et al. (2019) did also not support a causal role of life-long serum 25(OH)D reduction on depression [190]. 

Several RCTs in Asian patients with major depression suggest that vitamin D may play a role in the adjuvant treatment of depression. A recent RCT (published after the search period) with 78 elderly depressed patients [191], which was not included in the MA by Vellekatt [180], found a significant beneficial effect of vitamin D supplementation on depressive symptoms. However, all RCTs with depressed patients were conducted either in Iran or in China, and as 25(OH)D levels are influenced by sunlight exposure, replication in other populations is needed. 

In sum, the evidence reviewed does not support primary prevention of depression by vitamin D supplementation. Limited evidence suggests that vitamin D supplementation may be an efficacious adjunctive treatment for major depression, and a single RCT indicates that postpartum depression may also be alleviated. The quality of the reviewed SRs on vitamin D and depression is very heterogeneous. Several RCTs, including a Finnish study with 3000 unipolar depressive patients (clinicaltrials.gov identifier: NCT02521012) are underway to allow for a more definitive assessment of the possible antidepressant effects of vitamin D.

### 3.3. Autoimmune Diseases

#### 3.3.1. Multiple Sclerosis


**Background**


Multiple sclerosis (MS) is a demyelinating disease of the central nervous system and manifests as acute relapses and progressive disability. Autoreactive T-cells and B-cells are able to migrate through the blood–brain barrier and enter the brain, where they cause inflammation [192]. This persistent inflammation leads to loss of neuronal synaptic functions. Domains such as mobility, sensory perception, hand function, vision, fatigue, etc. are commonly affected [193]. Methods to quantify the progression of MS are, for instance, the expanded disability status scale (EDSS) [194], the annual relapse rate (ARR) or measurement of brain lesions by magnetic resonance imaging (MRI), using two different methods with distinct settings (T1 or T2–weighted sequences) [195].

MS mostly appears in adults in their late 20s or early 30s. Worldwide, an increase in the prevalence of MS has been reported over the past decades. However, to a large extent this could be explained by better diagnostic accuracy [196]. According to data from 2013, prevalence varies between countries and is highest in North America (140 per 100,000) and Europe (108 per 100,000) and lowest in Africa (2.1 per 100,000) and East Asia (2.2 per 100,000) [197]. Furthermore, mortality rate among MS patients is higher compared to the general population [198]. 

The etiology of MS is still unknown, but it is suggested that multiple factors, such as genetics, low vitamin D status, Epstein–Barr virus infection, obesity in childhood or adolescence, or smoking play a role in the development of MS [199]. 

Numerous studies have shown a latitude-dependent variation in MS incidence [200]. Hence, the duration and intensity of UV radiation and the resulting endogenous synthesis of vitamin D might influence the development of MS during life. Furthermore, several studies have examined the association between genetic factors and MS. As for other auto-immune diseases, genetic variants within the HLA-complex are among the strongest genetic risk factors. Also, non-HLA single nucleotide polymorphisms, influencing the immune system or the serum 25(OH)D level, have been discussed as potential risk factors [201,202].

Thus, an adequate vitamin D status reveals a possible approach for primary prevention of MS and also for modifying the natural course of the disease [203]. However, whether or not a low serum 25(OH)D level is causally linked to the onset and course of MS is still a matter of scientific debate. In this review, we therefore aimed to summarise data of SRs regarding a possible role of vitamin D in the prevention and therapy of MS.


**Results**


Overall, 11 records were included in this review [100,204,205,206,207,208,209,210,211,212,213] (Appendix A). We focused on publications that examined the association between vitamin D and the development of MS or clinical activity of MS. Appendix A provide information about the included studies and the assessed quality based on AMSTAR 2. Furthermore, three Mendelian randomisation studies were identified [214,215,216].


**Primary Prevention**


Regarding primary prevention, three SRs included data from prospective cohort studies [100,204,205]. According to AMSTAR 2, study quality was low for two SRs and high for one [204]. However, most of the studies were looking at MS patients and very few at non-diseased population groups. The most recent SR [205] included results from the Nurses’ Health Study as the first large prospective study that examined the risk of developing MS in a group of non-diseased women. The authors reported a 40% lower risk of developing MS in U.S. women who supplemented vitamin D, but failed to find an association for vitamin D intake by food consumption [217]. A prospective study examining the association between serum 25(OH)D concentrations and risk of MS in a cohort of US military personnel found an inverse association in whites, but not in blacks or Hispanics [218]. This paper was included in the SR provided by Autier et al. [100]. 

Observational studies remain limited by residual confounding and potential for reverse causation, particularly in MS where time of onset of pathophysiological processes remains known. Recent studies using Mendelian randomisation have provided some support for a causal role of low vitamin D status in the development of MS [214,215,216]. Mokry et al. were the first to explore the idea of a causal relationship between vitamin D status and risk of MS by defining an instrumental variable using serum 25(OH)D predicting SNPs (four SNPs explaining an important proportion of the population-level variance in 25(OH)D level) and applying the Mendelian randomisation approach [214]. Rhead et al. conducted a combined MA including two studies and using an instrumental variable consisting of three SNPs associated with serum 25(OH)D concentrations [215]. The authors found strong evidence that low serum 25(OH)D concentration is causally related to MS. The same working group was also able to confirm this suggested causal association for paediatric-onset MS by analysing three SNPs which together explain about 2.8% of the variance in 25(OH)D levels [216].


**Therapeutic or Adjuvant Vitamin D Supplementation in MS Patients**


Regarding clinical outcomes, eight publications that conducted SRs of mainly RCTs [204,205,207,209,210,211,212,213] focused on changes in the expanded disability status scale (EDSS). In summary, the studies found no significant therapeutic effect of vitamin D on EDSS and/or concluded that vitamin D supplementation had no effects on EDSS. The study quality according to AMSTAR 2 ranged from low to high.

Considering annualised relapse rate (ARR), researchers of four more recent studies [206,208,210,211,213] found that vitamin D did not reduce the number of relapses. 

Data on MRI outcomes in RCTs regarding new T1 or T2 brain lesions is inconsistent. Some RCTs hint at a potential clinical treatment effect in favour of vitamin D, while others do not report a change in T1 or T2 brain lesions compared to the placebo group [204,207,210,211,213].

The authors of the most recent publication of the Cochrane Collaboration concluded that available very low-quality evidence does not suggest benefits of vitamin D for important health outcomes in patients with a diagnosis of MS [210].


**Discussion and Conclusion**


The prevalence of MS varies geographically, namely that in areas increasingly distant from the equatorial region—and thus with less UV exposure to sunlight—higher numbers occur. However, such ecological data are often misleading and other study designs are necessary to clarify a causal role of vitamin D supply for the development of MS. Observational studies should be prospective in nature and capture the right time window when the disease starts to develop. With respect to the peak age of MS diagnoses in the 20s or early 30s, many available cohorts are not suited to address this question because most of the adult cohort studies start at a higher age. Thus, available studies are few and probably do not address the vulnerable phase important for predisposition of this disease. An exception may be cohort studies starting very early in life that also recorded paediatric-onset MS. 

The study quality of the included SRs as assessed by the AMSTAR 2 tool was heterogeneous (Appendix A). Based on the few prospective studies included in the identified SRs, no firm conclusion on the role of vitamin D status and the risk of MS in adults can be drawn. This is in agreement with the results of a recently (after the search period) published SR (including MA) focusing on vitamin D in early life and later risk of MS; three studies found inverse relationships while the remaining three found no significant associations [219].

The emerging use of Mendelian randomisation studies as a new statistical tool to better approach causality helps to overcome some limitations of observational studies [84]. Several preconditions need to be fulfilled in order to draw valid estimates from them, and violations of these conditions cannot be fully excluded. Interpretation of MR study results has to take such risks into account. Nonetheless, all three available MR studies on vitamin D status and the development of MS assume a causal association. These findings are judged as important since interventional studies for the primary prevention of MS are difficult to perform.

The currently available evidence does not support a beneficial effect of vitamin D supplementation (and thus a better vitamin D status) on the natural course of diagnosed MS. However, the underlying individual studies are of low quality as judged, e.g., by a Cochrane Collaboration group [210]. Most studies administered vitamin D_3_ or D_2_ orally in high/pharmacological doses. In the SOLAR study, for example, 232 MS patients received a daily dose of vitamin D_3_ (167 µg (6680 IU) for the first four weeks, 350 µg (14,000 IU) for the following 44 weeks) over a period of 48 weeks, and no significant change was observed for the primary endpoint, which was the percentage of patients with disease activity-free status [210]. Several ongoing studies (clinicaltrials.gov identifier: NCT03385356, NCT01490502, NCT01817166, and NCT01440062) with different study designs and characteristics of the MS patients (including baseline vitamin D status) will provide additional information to guide clinicians and MS patients. 

At the moment, however, MS patients seek guidance on the question of vitamin D supplementation and optimal vitamin D status. For example, the Multiple Sclerosis Society of Canada has developed recommendations that take into account both the hopes of patients for possible favourable effects on MS symptoms and development and possible negative effects through very high doses of supplementary vitamin D. All statements are based on the available scientific literature [220].

In conclusion, with respect to the results of few prospective cohort studies in early life and in adulthood, there is insufficient evidence to conclude an inverse association between vitamin D status and the incidence of MS. Recent Mendelian randomisation studies, however, indicate a causal relationship between vitamin D status, characterised by serum 25(OH)D concentrations, and the risk of developing MS. 

There are several SRs examining suggested effects of oral vitamin D administration on the natural course of MS. According to the most recent high-quality report, i.e., the 2018 Cochrane Report, the current conclusions are based on low-quality evidence and overall do not support a clear benefit of vitamin D supplementation on MS patients’ outcomes.

#### 3.3.2. Type 1 Diabetes Mellitus


**Background**


T1DM is an autoimmune disorder characterised by an almost complete defect in insulin secretion. Short-term symptoms are hyperglycaemia or hypoglycaemia, the latter being the result of insulin treatment. Well-known long-term consequences are micro- and macrovascular changes, retinopathy, nephropathy, or neuropathy [221]. T1DM is generally diagnosed early in childhood or adolescence; in many countries, its prevalence is annually increasing by estimated 3% with broad geographical differences. The highest prevalence rates are observed in Europe and North America/Caribbean [222]. Overall, T1DM accounts for about 5–10% of all diabetes cases [223]. 

The causes of T1DM are still unclear; presently, several environmental and genetic factors are being discussed. There is no doubt that polygenetic disorders, specifically based on variants in HLA-complex genes coding for immunoactive proteins, increase the risk of T1DM [224]. Potential environmental risk factors are, for instance, viral infections, or intoxications. 

Accumulating evidence has emerged for an association between vitamin D status and T1DM risk [225]. Modulated by activation of the VDR, vitamin D influences several metabolic pathways and affects the immune system. Interestingly, the VDR has been identified in the membrane of pancreatic islet cells, suggesting a potential role of vitamin D for glucose homoeostasis [30]. Consequently, vitamin D (supplementation) might retard the progression of T1DM, probably by stimulating insulin secretion in the pancreatic ß-cells, even after the onset of the disease, or by immunomodulatory effects. We therefore aimed to summarise recent knowledge with respect to the role of vitamin D in the prevention and therapy of T1DM.


**Results**


Overall, three records were included in this paper with heterogeneous study quality [226,227,228] (Appendix A). Appendix A provide information about the included studies and the assessed quality based on AMSTAR 2. No Mendelian randomisation studies were identified.


**Primary Prevention**


A MA including six case-control studies and two cohort studies (low quality according to AMSTAR 2) reported an inverse association between vitamin D supplementation in early life and risk of T1DM with an odds ratio of 0.71 (95% CI: 0.51–0.98) [227]. Subgroup analyses by study design showed that the pooled risk estimate for the case-control studies was similar to the overall one (OR = 0.68; 95% CI, 0.49–0.94); however, for the two cohort studies it was non-significant with a wide CI (relative risk = 0.62; 95% CI, 0.11–3.45). A further MA [226] reported that supplementation of vitamin D in infants is significantly associated with the risk of developing T1DM (four case-control studies; results of MA supported by one cohort study), though the quality according to AMSTAR 2 was low [226,229].

MAs of observational data indicate that maternal vitamin D intake during pregnancy possibly has no effect on the risk of T1DM in the offspring [226,227]. 

There were no SRs of RCTs available regarding vitamin D and the risk of T1DM.


**Therapeutic or Adjuvant Vitamin D Supplementation**


Gregoriou et al. conducted a SR of high quality (AMSTAR 2) including seven RCTs. They evaluated the effect of vitamin D_3_, calcitriol and its active analogues (i.e., alfacalcidol) on the daily insulin dose (DID) and the glycaemic indices [228]. While significant beneficial effects on DID, fasting C-peptide levels and stimulated C-peptide levels were observed after supplementation with alfacalcidol (0.25–0.5 µg per day) and vitamin D_3_ (50 µg per day (2000 IU) or 1.75 µg/kg body weight per day (70 IU/kg body weight)), no effects were seen after supplementation with calcitriol. Also, Antico et al. concluded that application of alfacalcidol significantly reduced the insulin requirement and protected the β-cell function compared to controls [226].


**Discussion and Conclusion**


Regarding primary prevention of T1DM, there are no RCT data on vitamin D supplementation available. MAs of observational data derived from case-control and a few cohort studies indicate that vitamin D supplementation in infancy significantly reduces the risk to T1DM later in life [226,227]. By contrast, maternal vitamin D intake during pregnancy is not associated with risk of T1DM in the offspring [226,227], and the few data on maternal or infant 25(OH)D concentrations and risk of T1DM in the offspring are inconsistent, showing either a protective effect or no association [230]. Epidemiological data on an increased risk of T1DM with increasing distance from the equator and the seasonal pattern with the highest risk of onset of T1DM in winter have been argued to support a protective role of vitamin D against T1DM [231], although ecological studies cannot provide information about causality. The study quality as assessed by the AMSTAR 2 tool [37] was heterogeneous (high and low quality) for the included SRs (Appendix A). There have been numerous genetic studies [232,233] mainly on SNPs of the VDR and T1DM, but these investigations produced inconsistent results and do not substantially contribute to our research question. Mendelian randomisation studies on 25(OH)D and T1DM are missing so far. 

Regarding vitamin D administration to patients with T1DM, the main findings of the only identified SR by Gregoriou et al. (2017) were significantly beneficial effects on DID, fasting C-peptide levels and stimulated C-peptide levels. These effects occurred after supplementation with alfacalcidol and vitamin D_3_, but not calcitriol [228]. While these findings are promising and there are also mechanistic data suggesting that vitamin D may protect against autoimmune diseases, it is premature to claim that vitamin D supplementation improves glucose homoeostasis or the natural course of T1DM. The above mentioned RCT data are clearly limited by a very low number of study participants and by multiple testing of various outcome measures of glucose homoeostasis, so that larger RCTs are required before drawing final conclusions. It is questionable whether a large long-term RCT on prevention of T1DM by vitamin D supplementation will ever be conducted, but further RCT data on effects of vitamin D supplementation in patients with T1DM can be awaited in the future (e.g., clinicaltrials.gov identifier: NCT03046927, NCT01724190, and NCT01029392.

## 4. Overall Discussion and Conclusion

The present umbrella review included results of 73 SRs (with or without MAs) regarding 25(OH)D status or vitamin D supplementation and various extraskeletal diseases. Main findings are summarised in Table 2. Additionally, Mendelian randomisation studies were identified to complement the results of the included SRs. 

Regarding primary prevention, observational data indicate an inverse association between vitamin D status and risk of ARI and depression, as well as dementia and cognitive decline, whereas data on asthma and MS and T1DM are inconclusive/insufficient. Notably, observational data are subject to unexplained confounding and the results are supported by MAs of RCTs only for ARI. However, for some other diseases such as COPD, MS, or T1DM, data from SRs of RCTs were simply not available, with the exception of depression, where no beneficial effect is suggested in the general population. 

With respect to possible effects on the course of the disease, SRs of RCTs suggest a beneficial impact of vitamin D in deficient patients with asthma and COPD, whereas no favourable effects have been reported in patients with ARI and MS. In patients with major depression and T1DM, respective studies are too scarce to draw a final conclusion.

Compared with the conclusions reported in previously published umbrella reviews [30,31], our evaluation considering more current studies leads to similar results. Consistent with our conclusion, both previous reviews (only focusing on MAs of RCTs) reported beneficial effects of vitamin D regarding the risk of respiratory tract infections and no effects of vitamin D supplementation on depression scores. Also, in accordance with our evaluation on the impact of vitamin D in patients with MS, Autier et al. (2017) reported no favourable effects [31]. A very recently published narrative review indicated a possible health benefit of vitamin D on COPD, ARI, as well as Alzheimer’s disease and other dementia, albeit based only on one or two recently published individual studies [234].

There are several limitations that have to be considered while interpreting the results of the included SRs in the present umbrella review. First, the issue of initial 25(OH)D status was not always adequately addressed. It is rather unlikely that individuals with an adequate vitamin D status will benefit from vitamin D supplementation. For several of the selected extraskeletal diseases (asthma, COPD, ARI) SRs of RCTs reported on a beneficial effect of vitamin D supplementation especially in persons with low baseline 25(OH)D concentrations (<25 nmol/l (10 ng/mL)), whereas in persons with adequate baseline 25(OH)D levels (>50 nmol/l (20 ng/mL)) often no effects were observed. In line with this, SRs of observational studies on several of the reviewed extraskeletal diseases (e.g., depression, dementia, and cognitive decline) support this finding concerning 25(OH)D status. Thus, it seems logical that large vitamin D RCTs with inclusion of individuals with adequate serum 25(OH)D levels failed to show significant effects [235].

Second, the mode, dose and frequency of supplementation varied widely, including low (10 µg (400 IU)) and high (100 µg (4000 IU)) daily doses and high intermittent bolus doses (3000 µg (120,000 IU)). In addition, different forms of vitamin D were administered (e.g., vitamin D_3_, or vitamin D_2_), and in some studies a multivitamin supplement was used or vitamin D was applied together with calcium. Ideally, vitamin D_3_ supplements at low daily doses and frequent intervals should be performed since this leads to a continuous elevation of 25(OH)D levels into the physiological range [126,236]. Third, the inconsistent definitions of endpoints (especially heterogeneous for the outcomes ARI and dementia) are an issue which might have influenced results. Fourth, duration of the individual studies varied widely (e.g., for COPD from six weeks to one year) as well as the studied population (from new-borns to elderly). Fifth, for some diseases such as COPD, dementia, depression, MS, and T1DM, the assessed quality of the SRs was very heterogeneous. 

Apart from these limitations, the additionally identified Mendelian randomisation studies only partially supported the results of the SRs. However, Mendelian randomisation studies are not without weaknesses because they are based on assumptions which are often only partly fulfilled: (i) single nucleotide polymorphisms (SNPs) are directly associated with the risk factor; (ii) SNPs are not the cause of known or unknown bias; and (iii) SNPs do not influence the risk of the disease through other factors. Moreover, it must be kept in mind that genetic variants explain only a small fraction of the 25(OH)D variance. Consequently, effects in subgroups at increased risk of either 25(OH)D deficiency or for the outcome may be overlooked.

To conclude, with the exception of beneficial vitamin D effects on the prevention of ARI and on the treatment of vitamin D deficient patients with asthma and COPD, available data do not clearly support beneficial vitamin D effects in preventing the reviewed extraskeletal diseases or affecting the course of the disease. Therefore, further high-quality RCTs are warranted. Ideally, these RCTs should be performed in patients with insufficient vitamin D status, and moderate daily vitamin D doses should be preferred. Also, geographical aspects and seasonal variations contributing to an altered sun-exposure must be considered. In addition, the trials should be designed with well-defined diseases or endpoints. Furthermore, they should be adequately powered to assess whether or not vitamin D is able to influence clinically relevant endpoints. Both efficacy and safety should be adequately considered.

The main strength of the present umbrella review is the comprehensive overview of systematically identified SRs (with or without MAs) of RCTs and cohort studies on vitamin D status and supplementation and the seven selected diseases, thereby updating the knowledge concerning the possible role of vitamin D for the development and course of these diseases. More importantly, we assessed the methodological quality of all included SRs with the AMSTAR 2 tool so that our conclusions are also based on the quality of the included studies. A limitation of this review is that we did not evaluate the quality of evidence of the studies and that limitations inherent to the original individual studies cannot be further reduced by our umbrella review.

As identified at clinicaltrial.gov, several RCTs are already underway that may provide additional information on the effect of vitamin D on the seven extraskeletal diseases investigated in this review. Besides, for diseases which develop over a longer time span, such as dementia, or a certain period of life (e.g., MS diagnosis in early adulthood), vitamin D effects may be difficult to assess in RCTs. Therefore, well-designed longitudinal observational studies could be extremely helpful to provide further information. 

Generally speaking, ingested or endogenously produced vitamin D doses of 20 µg (800 IU) are considered safe, and can significantly contribute to attain circulating 25(OH)D concentrations of at least 50 nmol/l (20 ng/mL). However, if supplementation is necessary (e.g., due to lack of endogenous synthesis), continuous daily doses (in the amount of the recommended daily intake level of 10–20 µg (400–800 IU) should be applied instead of high bolus doses [2,3,4,5,237]. 

## Figures and Tables

**Figure 1 nutrients-12-00969-f001:**
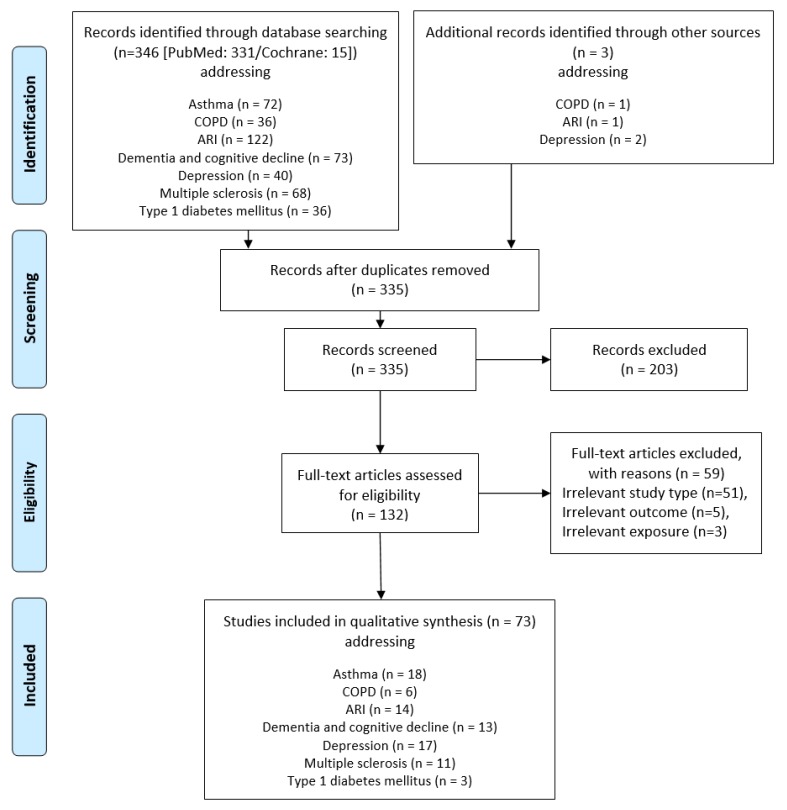
PRISMA flow diagram—all outcomes.

**Table 1 nutrients-12-00969-t001:** Inclusion and exclusion criteria.

Inclusion Criteria	Exclusion Criteria
Published in English or German language	Articles that were only available as conference proceedings or in abstract form
Published since 2010	Top athletes
General population	SRs based on case-control or cross-sectional studies only
Study type: SRs (with or without MAs) of at least two cohort studies and RCTs^1^	Umbrella or narrative reviews
MAs and SRs including studies in infants, children, and adolescents or in pregnant or breastfeeding women or birth cohort studies were only eligible for specific outcomes (asthma, depression, T1DM, ARI)	Association between 25(OH)D status or the effect of vitamin D supplementation on outcome not evaluated
Exposure: 25(OH)D status or vitamin D supplemen-tation (vitamin D, vitamin D_3_, vitamin D_2_, calcitriol, alfacalcidol)^2^	Duplicate/dated publication on the same exposure and outcome
Outcome: ARI, asthma, COPD, dementia and cognitive impairment, depression, T1DM, MS	

^1^ It was tolerated that some SRs included case-control studies or cross-sectional studies besides RCTs and/or cohort studies. ^2^ Different forms of vitamin D supplementation were eligible, since there are hardly any SRs including studies with vitamin D_2_ or D_3_ supplementation alone.

**Table 2 nutrients-12-00969-t002:** Summarised results of the included SRs on vitamin D and the seven selected extraskeletal diseases. (ARI, acute respiratory tract infections; COPD, chronic obstructive pulmonary disease; MS, multiple sclerosis; RCTs, randomised controlled trials; SRs, systematic reviews; T1DM, type 1 diabetes mellitus).

	SRs of	Effects of Vitamin D in Primary Prevention	Effects of Vitamin D in Patients
**Asthma**	Observational studies	-	no data
	RCTs	-	+^1,2^
**COPD**	Observational studies	no data	-
	RCTs	no data	+^1^
**ARI**	Observational studies	+^3^	no data
	RCTs	+^1^	o
**Dementia and cognitive decline**	Observational studies	+^1,4^	no data
	RCTs	-	no data
**Depression**	Observational studies	+^1,4^	no data
	RCTs	o (general)– (postpartum)	(major depression)
**MS**	Observational studies	-	no data
	RCTs	no data	o
**T1DM**	Observational studies	-	no data
	RCTs	no data	-

+ Beneficial effect suggested—final conclusion is not possible due to heterogeneity of data or limited data. - No clear statement possible due to inconclusive/insufficient data. o No beneficial effect suggested. ^1^ Especially or only in patients with circulating 25(OH)D concentrations <25 nmol/l (10 ng/mL). ^2^ Most results are based on SRs in children. ^3^ Results are based on SRs in adults. ^4^ Effects mainly seen in older persons.

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
