# Peer review of "Role of Vitamin D in Preventing and Treating Selected Extraskeletal Diseases—An Umbrella Review"

_nutrients, 2020, doi:10.3390/nu12040969_

Round 1
Reviewer 1 Report
- Suggestion for Supplemental Tables containing the description of SRs and MAs. The most right column contains the evaluation of quality assessed by the AMSTAR 2 tool, however, the quality status is not stated for some MAs. In the main manuscript text, there is an explanation that some MAs are PID MAs and for those MAs the AMSTAR 2 tool could not be aplied. It's OK, but it would be better if in such cases the right column of the Supplemental Tables contains a simple statement, e.g. "Not eveluated", "Not suitable", or "Not to be determined".
- Manuscript text, line 315: it seems that there is a missing number regarding the CI. A range of 95% CI is expected, i.e. "X, -0.70". Please provide the missing lower bound (X) of CI.
Reviewer 2 Report
The paper for the most part is well-written and the authors appropriately revised the manuscript.
- The authors should emphasize the limitations of the study realized
- The 2.2. Search strategy and eligibility criteria section at page 3 of the manuscript, is very important. The authors should explain in more detail the inclusion/exclusion criteria. In this sense, Table I should be more extensive. The table helps easily understand for readers
Reviewer 3 Report
The present article aims to review and update the current knowledge about the potential role of vitamin D in prevention and therapy of respiratory, autoimmune, neuro-degenerative and mental disorders.
A few changes are needed, as follows:
Introduction, lines 79-80: Please add also that in Rheumatoid Arthritis, low serum levels of vitamin D were associated with disease activity, increased insulin resistance, and endothelial dysfunction (Caraba A. et al. Vitamin D Status, Disease Activity, and Endothelial Dysfunction in Early Rheumatoid Arthritis Patients. Dis Markers. 2017;2017:5241012. doi: 10.1155/2017/5241012).
Related to asthma, please include a few words about the influence of passive smoking on vitamin D level (Chinellato I, et al. Correlation Between Vitamin D Serum Levels and Passive Smoking Exposure in Children With Asthma. Allergy Asthma Proc 2018. PMID 29669660).
Related to COPD, please mention the link between smoking (blood serum concentrations of cotinine) and vitamin D (Manavi KR, et al. History of tobacco, vitamin D and women. Int J Vitam Nutr Res. 2020 Feb 24:1-6).
Page 12, lines 517-519: Please mention also several other effects of vitamin D: the antiatherogenic effects, improvement of endothelial function, arterial elasticity and metabolic profile and inhibition of the renin-angiotensin-aldosterone system (Mozos I et al. Crosstalk between Vitamins A, B12, D, K, C, and E Status and Arterial Stiffness. Dis Markers. 2017;2017:8784971. doi: 10.1155/2017/8784971).
Reviewer 4 Report
This umbrella review represents a useful, albeit incomplete, contribution to the clinical literature on vitamin D. While the introductory section to the whole review needs some revisions, the introductory sections to each disease are sufficiently comprehensive. The analyses of data quality represent a strong point of the study. However, there are some deficiencies in other areas.
My specific comments are as follows:
- The Introduction starting on line 47 needs some revisions and expansion. While it indicates that vitamin D can be obtained from sufficient cutaneous UV exposure, it does not formally introduce the notion of vitamin D winter or that fact that no cutaneous vitamin D synthesis occurs above 45 degrees latitude (many major population centers of Europe) for more than 6 months of the year (at sea level). It should also introduce the notion of sun-avoidance (remarkably, some of the highest levels of vitamin D deficiency can be found in very sunny Saudi Arabia and the Emirates because of a combination of conservative dress and sun avoidance).
In addition, it does an inadequate job of defining vitamin D deficiency. Apparently, IOM guidelines for defining vitamin D sufficiency were used. There was considerable backlash from the scientific community upon release of these guidelines, including a rebuttal rapidly published by the US Endocrine Society. As a result, it is not correct to state that 50nM 25OHD is “generally considered” desirable to support bone health (line 68). Most people, including the Endocrine Society, consider 75-80nM 25OHD to correspond to vitamin D sufficiency. This is important because it expands the populations considered to be vitamin D deficient. One of the reasons that such an umbrella analysis is helpful to the clinical/research community is because is vitamin D deficiency is widespread. In this regard, specific examples of studies revealing the extent of vitamin D deficiency should be described (rather than just stating that they have been reported (line 69). The work of Cashman and colleagues would be helpful from a European perspective.
Finally, while everyone understands RCTs, it would be helpful to define at the outset what an MR study is and to what degree the variants in vitamin D metabolic genes used in MR studies collectively contribute to variations in 25OHD levels.
- The Martineau results (ref. 93) providing evidence that bolus dosing is not efficacious are of significance over and above their relevance to ARTs. Evidence that bolus dosing is not efficacious should be introduced prior to the analysis of individual diseases. More importantly, within each section it should be stated whether each RCT discussed used bolus dosing or daily/weekly dosing.
- The umbrella has a hole in it. Cancer should be included in the study. There is substantial laboratory and preclinical evidence for cancer-preventive actions of vitamin D signaling. In the clinic, the recent results of clinical trials of Manson and collaborators at Harvard Medical School and the subsequent MA by Giovannucci and Manson and collaborators are important in this regard.
Round 2
Reviewer 4 Report
The authors have made an excellent job of responding to reviewers' concerns.